# Severe Oral Mucositis in Pediatric Cancer Patients: Survival Analysis and Predictive Factors

**DOI:** 10.3390/ijerph17041235

**Published:** 2020-02-14

**Authors:** Lecidamia Cristina Leite Damascena, Nyellisonn Nando Nóbrega de Lucena, Isabella Lima Arrais Ribeiro, Tarciana Liberal Pereira, Luiz Medeiros Araújo Lima-Filho, Ana Maria Gondim Valença

**Affiliations:** 1Departament of Statistics, Federal University of Paraíba (UFPB), João Pessoa, PB 58051-900, Brazil; nyellisonobrega@hotmail.com (N.N.N.d.L.); tarcianalp@gmail.com (T.L.P.); luiz@de.ufpb.br (L.M.A.L.-F.); ana.valenca@ccs.ufpb.br (A.M.G.V.); 2Departament of Public Health, Ribeirão Preto Medical School, University of São Paulo, Ribeirão Preto, SP 14049-900, Brazil; ilaribeiro@hcrp.usp.br

**Keywords:** oral mucositis, cancer, child, survival analysis

## Abstract

This paper investigates the occurrence of severe oral mucositis and associated factors in blood and solid cancer pediatric patients subjected to cancer treatment, using a survival analysis. A longitudinal, descriptive, observational and inductive study of 142 pediatric patients aged from 0 to 19 years was conducted from 2013 to 2017. Data were collected using a form to record the sociodemographic characteristics and health-related aspects of patients and the modified Oral Assessment Guide (OAG). Survival analysis was performed using the Kaplan–Meier method and Cox semiparametric model. The median times to occurrence of severe oral mucositis were 35.3 and 77.1 days for patients with hematologic malignancies and solid tumors, respectively. The Cox model showed that white cell changes and platelet counts as well as the use of natural chemotherapeutic agents are risk factors for the occurrence of oral mucositis among patients with hematologic malignancies. Nonetheless, among patients with solid tumors, the occurrence of oral mucositis was associated with female sex, mixed ethnicity, the presence of metastasis, abnormal creatinine levels, a combination of chemotherapy, radiotherapy, and surgery, and the administration of chemotherapeutic agents included in the miscellaneous group. The time to occurrence of severe oral mucositis and its associated factors varied according to cancer type.

## 1. Introduction

Cancer is considered a rare disease among children and adolescents, corresponding to 1% of all neoplasms in this group, with an estimated rate of one case for every 408 individuals under 15 years old and one case for every 285 individuals under 20 years old [1]. Despite an approximate 80% increase in the survival rate, cancer is still the second leading cause of death for the pediatric population, following only accidents [1].

Cancer treatment may cause several side effects, including severe oral mucositis (SOM). Oral mucositis is an inflammation of the oral mucosa that causes pain, ulceration, and bleeding [2]. The incidence of SOM has been associated with chemotherapy regimens, including the amount and time of administration of doses [3].

Depending on its severity, SOM can become a disabling condition, as it hinders nutrition, requires the administration of powerful analgesics, increases the length of hospital stay and treatment costs, and above all, impairs the quality of life of patients [4,5]. In addition, SOM may be a reason for the partial or definitive discontinuation of cancer treatment [6], thus compromising chemotherapy effectiveness and resulting in a higher risk of cancer cell proliferation and poorer disease control.

Children and adolescents are more prone to develop SOM, and younger children have a higher odds of occurrence of chemotherapy-induced SOM. Approximately 40% of patients receiving chemotherapy manifest side effects affecting the mouth; this rate increases to more than 90% among children under 12 years old [6,7,8].

SOM may be a determinant for the discontinuation of cancer treatment because partial or total interruption is required in severe cases, with a consequent negative impact on patients because from the clinical perspective, pediatric tumors grow faster and are more invasive [7,8]. Therefore, an analysis of the time from the onset of cancer treatment to the occurrence of SOM is relevant, as it might elucidate how this condition affects patient prognosis.

Given the above information, the aim of the present study was to determine the variables that influence the time to occurrence of SOM among pediatric cancer patients using a survival analysis.

## 2. Materials and Methods

### 2.1. Study Design

The study was approved by the research ethics committee of the Center of Health Sciences, Federal University of Paraiba (Universidade Federal da Paraíba—UFPB), under CAAE (Presentation Certificate for Ethical Appreciation) No. 63746316.0.0000.5188. The consent of participants aged 18 years and over was obtained through the consent form and for participants under 18 years of age, the consent of a parent or legal guardian was obtained. Additionally, all methods were performed according to relevant guidelines and regulations.

The present longitudinal and observational study sought to provide an explanation for the time interval to the occurrence of SOM among pediatric patients receiving cancer treatment. The sample comprised 142 children aged 0 to 19 years who met the inclusion criteria (first cancer diagnosis, not having started treatment, parental consent, being cooperative during application of the Oral Assessment Guide (OAG)) and were being treated at the pediatric unit of Napoleão Laureano Hospital, a referral center for cancer treatment in João Pessoa, Paraiba, Northeastern Brazil.

Data were collected from 2013 to 2017 using a form designed to gather information on sociodemographic characteristics and aspects related to the state of health of patients and the modified OAG [9]. The OAG comprises eight items, namely: voice, swallowing, lips, tongue, saliva, labial/palate mucosa, labial mucosa, and gingiva. Each item is assigned a score from 1 to 3, where 1 represents the normal state, 2 indicates slight to moderate abnormalities, and 3 indicates SOM. Data were collected by calibrated examiners (Cohen’s Kappa > 0.7) after the diagnosis was established. The observation lasted 10 weeks, beginning immediately after the medical staff initiated anticancer treatment.

Given that childhood cancer is a condition that is rare and difficult to follow up, study participants were recruited through convenience sampling for longitudinal and observational analysis. All children assisted from 2013 to 2017 whose mothers agreed to participate in the survey were included, totaling a sample size of 142 children. A posteriori analysis indicated statistical power of 77%, considering an alpha error of 5% and *n* = 142.

### 2.2. Statistical Analysis

Collected data were analyzed through descriptive statistics to determine the mean, standard deviation, and frequencies, and inferential statistics involved non- and semiparametric survival analysis methods, considering the occurrence of SOM as the event of interest. Cases that did not exhibit the event of interest (i.e., data censored) were the patients who did not develop mucositis during the 10-week follow-up period or who died during this period without exhibiting the event of interest.

A nonparametric statistic, the Kaplan–Meier estimator, was used to analyze the time from onset of treatment to the occurrence of SOM. Survival among groups of interest was compared using the log-rank test. The final regression model was obtained with the Cox semiparametric model; the model fit and adequacy were investigated using Martingale, deviance, and standardized Schoenfeld residuals and the proportionality test. All residual analyses indicated that the final model fit well to the data. The significance level was set to 5%. The dataset was analyzed with the free software R version 3.2.4 (The R Project for Statistical Computing, Free Software Foundation, Auckland, New Zealand).

## 3. Results

A total of 142 pediatric cancer patients were included in our study. Table 1 presents the sociodemographic characteristics of the sample. Most patients were males (*n* = 73; 51.4%) and identified as mixed race (“pardo”) (*n* = 73; 51.4%). There was a higher incidence of hematologic malignant (*n* = 80; 56.3%), and acute lymphoid leukemia was the predominant cancer type (*n* = 55; 38.8%).

Most of the pediatric cancer patients did not experience an occurrence of metastasis (*n* = 129; 90.1%), and death occurred in 13.4% of patients (*n* = 19).

Chemotherapy was the most frequently used treatment modality, which was reported by 69% (*n* = 98) of the individuals. The class of drugs used in the treatment regimens were antimetabolites (*n* = 63; 44.4%).

In addition, normal leukocytes count, platelets count, and creatinine level were the most commonly observed findings.

The survival analysis showed that 66.2% (*n* = 94) of the 142 included patients developed the event of interest (SOM) at some point during treatment. The occurrence of oral mucositis was 55.32% (*n* = 52) and 44.68% (*n* = 42) among patients with hematologic and solid cancer, respectively. For the patients with hematologic malignancies, the time to occurrence of SOM varied from 3 to 229 days, with a median of 35.3 days, i.e., the time for 50% of the participants to develop SOM. For patients with solid tumors, the time to occurrence of SOM varied from 1 to 149 days, with a median of 77.1 days. The Peto test, which verifies the difference in the survival curves of two groups, was performed to evaluate a potencial divergence in time of appearance of oral mucositis in both types of tumor. The *p*-value = 0.008 obtained in the test confirm the difference. Figure 1 and Figure 2 depict Kaplan–Meier curves for both types of cancer with the corresponding 95% confidence intervals.

Next, the Cox regression model was used to explain the risk of occurrence of SOM among patients with hematologic malignancies or solid tumors. Thus, the relative risk (RR) was calculated for each variable included in the final model fit for each patient group. The results are described in Table 2 and Table 3.

## 4. Discussion

The present study is the first to analyze the time to occurrence of SOM in pediatric patients with blood or solid cancers using survival analysis. In addition, it shows the influence of different variables on the occurrence of SOM according to the cancer type. Some of these variables were previously reported in the literature, whereas others were unknown.

Studies have shown that the incidence of oral mucositis differs according to the type of cancer and treatment regimen, suggesting that it is more frequent in those with hematological malignancies compared with patients suffering solid tumors [10,11,12,13]. Based on the occurrence of oral mucositis is different in these groups of tumors [10], in the present study, the patients were divided into two groups—hematologic malignancies and solid tumors. This classification is common in the literature: hematological malignancies and solid tumors. The guidelines of Multinational Association of Supportive Care in Cancer (MASCC) and European Society For Medical Oncology (ESMO) about hematological malignancies and their studies to determine predictive factors for complication in cancer patients include leukemias and lymphomas in the same group.

Survival analysis showed that the median time to occurrence of SOM differed between patients with hematologic malignancies and solid tumors. In another study, the mean time to occurrence of oral mucositis was 6 weeks for patients with acute leukemia, 1 week for patients with lymphoma, and 3 weeks for patients with solid tumors [10]. These results differ from those of the present study, in which the time to occurrence of SOM was shorter for patients with hematologic malignancies than for those with solid tumors. The median time to occurrence of SOM among patients with solid tumors was approximately twice as long as that for patients with hematologic malignancies. However, it is important to note that the measure of the central tendency used in these studies was different. So, the comparison between them would not be reliable and should be stated with caution.

The division of the patients in two groups might have influenced the results relative to the time to occurrence of SOM. In the aforementioned study, this time was longer for patients with leukemia and shorter for those with lymphoma. These are two varieties of blood cancer, and the fact that they were included in the same category in the present study might have resulted in a shorter overall time to occurrence of SOM.

Patients with hematologic malignancies develop oral complications more frequently than those with other types of cancer [12,14]. A study performed in India found a higher incidence of oral mucositis among patients with blood cancer [12]. In the present study, the frequency of oral mucositis was also higher among patients with hematologic malignancies. One study found that patients with acute leukemia exhibited a higher risk of oral mucositis, high concentrations of pro- and anti-inflammatory cytokines, and low plasma pro-LL-37 levels (a protein considered to have a protective role in oral health), findings which suggest that these factors might influence the development of oral mucositis in this patient population [10].

The Cox model fit for patients with blood cancer showed that a white blood cell count less than 9500/mm^3^, a platelet count greater than 450,000/mm^3^, and use of natural chemotherapeutic agents increased the risk of occurrence of SOM. Neutropenia and lymphopenia were previously associated with the occurrence of oral mucositis [15,16]. According to reports in the literature, the risk of oral mucositis is 3.08 times higher for patients with low neutrophil counts than for patients without this condition [15]; this value is similar to that found in the present study (3.22).

Neutropenia and lymphopenia are common findings among patients with hematologic malignancies and might also develop as an adverse effect of treatment [11,16]. According to one study, a reduction in neutrophils count can result in inefficient protection against damage to the oral mucosa and can impair epithelial cell proliferation [17]. The association between tumor-induced inflammatory response and cytolytic action of treatment might influence the development of oral mucositis [18].

A reduction in leukocytes count impairs the body’s defenses, as these cells are responsible for this function. Some evidence indicates an association between oral viruses, especially the herpes simplex virus, and candidiasis and the development of oral mucositis [19]. Neutropenic patients are at higher risk of bacterial colonization of the damaged epithelium, leading to inflammation secondary to increased levels of pro-inflammatory cytokines in the mucosa, which aggravates oral mucositis [16].

In the present study, an increase in the platelet count above the reference level was identified as a risk factor for the occurrence of SOM. This finding is controversial in the literature because, according to some reports, the occurrence of oral mucositis is associated with thrombocytopenia [7,20,21]. A possible explanation for this discrepancy is that some patients with thrombocytopenia received platelet concentrate transfusions to help them recover a normal platelet count. The transfusions might have been performed at the time of myelosuppression, when the body’s defenses are impaired, and this situation favors the occurrence of several side effects, including SOM.

The administration of natural chemotherapeutic agents also acted as a risk factor for the occurrence of SOM. Some of the agents in this class (daunorubicin, doxorubicin, vincristine, and etoposide) are included in the GBTLI-99 protocol for the treatment of acute lymphoblastic leukemia in patients with a low or high risk of recurrence [22], and this was the most frequent disease in the analyzed sample. Doxorubicin, paclitaxel, and etoposide, all natural chemotherapeutic agents, are considered to be stomatotoxic [9].

Vincristine, an alkaloid obtained from rose periwinkle that has antimitotic activity and is classified as a natural chemotherapeutic agent, was related to the occurrence of oral pain combined with ulceration [14]. Etoposide is excreted in saliva, which increases its oral toxicity [23]. Other natural chemotherapeutic agents are included in protocols for other varieties of blood cancer, such as acute myeloid leukemia and non-Hodgkin lymphoma [24,25].

Regarding patients with solid tumors, the Cox semiparametric model showed that female sex; mixed ethnicity; the presence of metastasis; a creatinine concentration of 0.4 to 1.3 mg/dL; a combination of chemotherapy, surgery, and radiotherapy; and the administration of chemotherapy agents classified as miscellaneous acted as risk factors for the occurrence of SOM. It is worth noting that patients who received chemotherapy combined with radiotherapy and surgery exhibited an approximately six times higher risk of SOM than patients subjected to other treatments.

The occurrence of SOM is dependent on factors related to cancer treatment and individual patient factors. Female sex was a risk factor for the occurrence of SOM. Other studies have also reported this relationship but have not provided any explanation for it [18,26,27]. A study that compared men and women found that female sex might be a relevant risk factor for the occurrence of oral mucositis induced by high-dose chemotherapy [28].

Mixed ethnicity was considered a risk factor for the occurrence of SOM among patients with solid tumors. The fact that the population of Brazil is predominantly mixed [29] might have influenced this finding. A study conducted in Brazil found a higher number of cases of solid tumors among nonwhite children than in white children [30]. In some European countries, central nervous system cancer is more frequent among white children [31]. In the United States, the incidence of malignant tumors is higher among white children; however, the mortality rate is the same for black children due to their lower survival rate [1].

Racial and ethnic disparities in cancer survival have been reported and might be associated with the pathobiology of the disease due to the impact of genetic polymorphisms on the metabolism of chemotherapeutic agents [1]. These variations might influence particular aspects of SOM occurrence in various racial groups.

The presence of metastasis acted as a risk factor for the occurrence of SOM among patients with solid tumors. Metastasis occurs when cancer cells migrate to other body sites where they give rise to new tumors. Metastasis typically requires more aggressive treatment than the primary tumor. In many cases, oral mucositis developed as nonhematological chemotoxicity after treatment for metastasis [32,33].

Changes in kidney function may also be associated with the occurrence of SOM. The creatinine reference values used in the present study were established for pediatric patients by the hospital laboratory, and the normal range was considered to be 0.4 to 1.3 mg/dL. Small variations in the creatinine level, even within the normal range, might have effects such as changes in kidney function [34,35,36].

Many chemotherapeutic agents—such as epirubicin, cisplatin, paclitaxel, cyclophosphamide, and melphalan—are nephrotoxic and increase the serum creatinine and urea levels [37,38]. Thus, the administration of drugs that alter the kidney function impairs their elimination from the body, with a consequent increase in their toxicity, which might facilitate the development of SOM.

A study that analyzed 666 patients with solid tumors found a high prevalence of an abnormal glomerular filtration rate [39]. In another study conducted on patients with kidney failure receiving melphalan—a stomatotoxic drug used for treatment of multiple myeloma—the severity of oral mucositis increased with high-dose chemotherapy, as drug elimination was impaired due to the patients’ abnormal kidney function [37]. Delayed methotrexate elimination led to increased creatinine levels and oral mucositis as a consequence of its toxicity [36]. These agents were administered to some of the patients analyzed in the present study.

One further risk factor for the occurrence of oral mucositis was the combination of chemotherapy, radiotherapy, and surgery. Chemotherapy and radiotherapy alone cause several side effects in the mouth, such as oral mucositis. The incidence of oral mucositis varies according to the treatment administered. Regarding chemotherapy, the severity of oral mucositis depends on the cytotoxic agents used and the strength of the therapeutic regimen. In the case of radiotherapy, the severity of oral mucositis is determined by the cumulative dose [40].

In general, all three therapeutic options—surgery, chemotherapy, and radiotherapy—are used alone or in combination for head and neck cancer patients [41,42]. The combination of all three options might make patients more vulnerable and thus increase the risk of complications such as SOM.

Cisplatin, which was included in the miscellaneous class of chemotherapeutic agents, is used for the treatment of some solid tumors, such as those affecting the bladder, ovaries, and testicles, and causes blood and kidney disorders [36], which might influence the occurrence of SOM.

The difficulty of measuring the time to occurrence of SOM in outpatients might impair the generalization of the results of the present study and thus represents one of its limitations. Another limitation derives from the descriptions given in most studies [8,11,12,16], which have addressed oral mucositis instead of SOM, therefore hindering the comparison of results.

Another limitation is the heterogeneity of the sample concerning the types of cancer included in the hematological malignancies and solid tumors groups, which require differentiated treatment regimens. Analysis considering a single type of cancer would allow more accurate information, reducing the possibility of compromising the results obtained. However, the strongest evidence regarding the onset, progression, and risk factors for oral mucositis comes from systematic reviews in which the findings are presented mainly considering the groups of tumors (hematological and solid)—not a specific tumor type [43,44,45].

These limitations notwithstanding, the results of the present study contribute to improving the understanding of the occurrence of SOM among children and adolescents subjected to cancer treatment. The findings indicate the need for differential attention for patients with hematological malignancies or solid tumors, as well as for further studies and strategies for the control and prevention of this condition.

## 5. Conclusions

The time to occurrence of SOM varied between groups according to the cancer type and was shorter for the patients with hematological malignancies than for patients with solid tumors.

The final models fit for patients with blood cancer and solid tumors showed that different variables influenced the development of SOM. These findings highlight the relevance of the present study, as it indicates the existence of differences in the occurrence of SOM for various cancer types.

## Figures and Tables

**Figure 1 ijerph-17-01235-f001:**
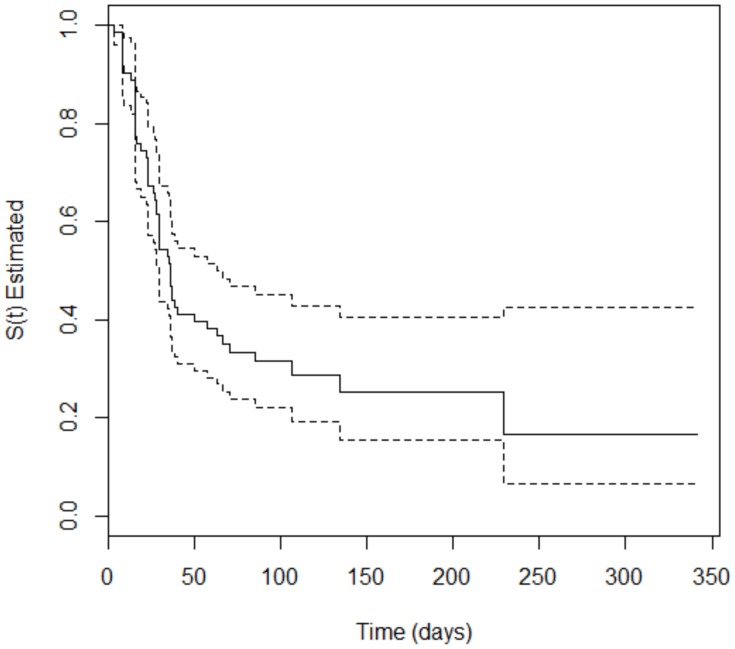
Kaplan–Meier curve (solid line) with 95% confidence interval (dotted line) corresponding to patients with hematologic malignancies.

**Figure 2 ijerph-17-01235-f002:**
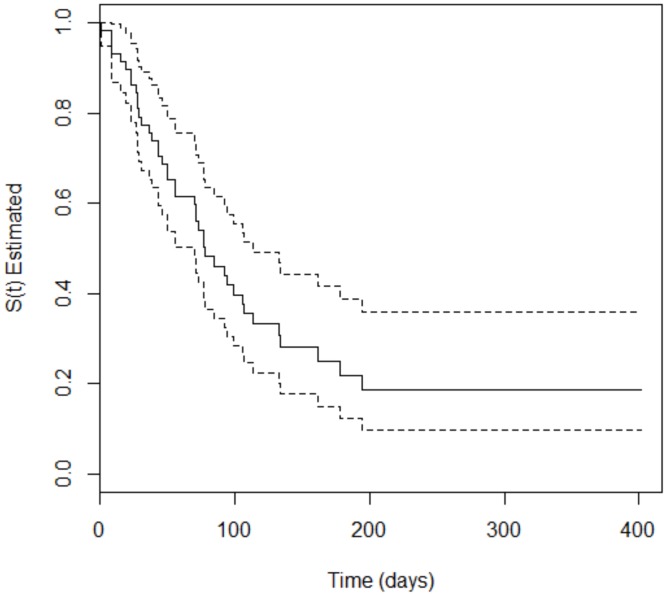
Kaplan–Meier curve (solid line) with 95% confidence interval (dotted line) corresponding to patients with solid tumors.

**Table 1 ijerph-17-01235-t001:** Distribution of absolute and relative frequencies of sociodemographic variables corresponding to pediatric cancer patients receiving cancer treatment at a referral hospital, João Pessoa, 2013–2017.

Variable	Relative Frequency %	Absolute Frequency *n*
**Sex**		
Male	51.4	73
Female	48.6	69
**Ethnicity**		
Mixed	51.4	73
White	33.1	47
Black	14.8	21
Indigenous	0.7	1
**Cancer type**		
Hematologic malignant	56.3	80
Solid tumor	43.7	62
**Disease**		
Acute lymphoblastic leukemia	38.8	55
Wilms’ tumor	14.1	20
Osteosarcoma	13.4	19
Non-Hodgkin lymphoma	9.2	13
Acute myeloid leukemia	6.3	09
Embryonal rhabdomyosarcoma	4.2	06
Adenocarcinoma	2.8	04
Hodgkin lymphoma	2.1	03
Neuroblastoma	2.1	03
Brain stem tumor	2.1	03
Bladder tumor	0.7	01
Spinal tumor	0.7	01
Germ cell tumor	0.7	01
Melanoma	0.7	01
Lymphoepithelioma	0.7	01
Alveolar soft part sarcoma	0.7	01
Synovial sarcoma	0.7	01
**Metastasis**		
No	90.1	128
Yes	9.9	14
**Death**		
No	86.6	123
Yes	13.4	19
**Treatment**		
Chemotherapy	69.0	98
Chemotherapy + surgery	24.0	34
Chemotherapy + radiotherapy	3.5	5
Chemotherapy + radiotherapy + surgery	3.5	5
**Class 1—Alkylating agents**		
No	83.1	118
Yes	16.9	24
**Class 2—Antimetabolites**		
No	55.6	79
Yes	44.4	63
**Class 3—Natural agents**		
Yes	58.5	83
No	41.5	59
**Class 4—Miscellaneous**		
No	83.1	118
Yes	16.9	24
**Bone marrow transplantation**		
No	97.2	138
Yes	2.8	4
**Limb amputation**		
No	73.2	104
Yes	26.8	38
**Leukocytes**		
3400 to 9500/mm^3^	48.6	69
Less than 3400/mm^3^	33.8	48
Greater than 9500/mm^3^	17.6	25
**Platelets**		
150,000 to 450,000/mm^3^	52.8	75
Less than 150,000/mm^3^	26.1	37
Greater than 450,000/mm^3^	21.1	30
**Creatinine**		
0.4 to 1.3 mg/dL	81.7	116
Less than 0.4 mg/dL	17.6	25
Greater than 1.3 mg/dL	0.7	1

**Table 2 ijerph-17-01235-t002:** Estimated parameters and relative risk (RR) corresponding to the final model for patients with hematologic malignancies treated at Napoleão Laureano Hospital, 2013–2017.

Hematologic Malignancies						
**Covariables**	**Risk Category**	**Estimate**	**Standard Error**	***p*-Value**	**RR**	**95% CI (RR)**
**Leukocytes**	Greater than 9500/mm^3^	−1.169	0.513	0.023	0.311	0.114–0.849
**Platelets**	Greater than 450,000/mm^3^	0.772	0.357	0.031	2.164	1.075–4.357
**Class 3—Natural agents**	Use of natural chemotherapeutic agents	0.673	0.299	0.025	1.961	1.089–3.528

RR: relative risck.

**Table 3 ijerph-17-01235-t003:** Estimated of parameters and relative risk (RR) corresponding to the final model for patients with solid tumors treated at Napoleão Laureano Hospital, 2013–2017.

Solid Tumors						
**Covariable**	**Risk Category**	**Estimate**	**Standard Error**	***p*-Value**	**RR**	**95% CI (RR)**
**Sex**	Female	1.033	0.392	0.008	2.809	1.304–6.052
**Ethnicity**	Nonmixed	−0.768	0.391	0.049	0.464	0.216–0.997
**Metastasis**	With metastasis	1.129	0.458	0.014	3.092	1.259–7.592
**Creatinine**	0.4 to 1.3 mg/dL	1.359	0.619	0.009	3.892	1.157–13.095
**Treatment**	Chemotherapy + surgery + radiotherapy	1.829	0.582	0.002	6.226	1.989–19.483
**Chemotherapeutic agent class**	Miscellaneous	1.063	0.407	0.028	2.890	1.303–6.432

RR: relative risck.

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
