# Peer review of "Severe Oral Mucositis in Pediatric Cancer Patients: Survival Analysis and Predictive Factors"

_ijerph, 2020, doi:10.3390/ijerph17041235_

Round 1

Reviewer 1 Report

Dear authors,

  I read your manuscript with great interest.

The paper il generally well written and the topic is interesting nevertheless I suggest you to make some changes before continuing its editorial process:

Line 73: Please change the number 9 that in superscript with square brackets;

Line 73: Regarding the 8 items of OAG please explain better to which areas they refer;

Line 75: Please write the correct definition ad "Cohen's Kappa";

Statistical analysis: Please provide a power analysis in which you explain how you decided to enroll a specific number of patients;

Line 80-82: Please rephrase the sentence as it is not clear;

Line 131-133: Why did you choose not to consider separately leukemias and lymphomas? I think this could really affect your results.

Line 134-135: Please add appropriate reference about the multidisciplinary and crucial role of dentists in helping the diagnosis of hematological head and neck malignancies (for your convenience: Patini et al. Early-stage diffuse large B-cell lymphoma of the submental region: a case report and review of the literature. Oral Surg. 2017;10:56-60).

Line 205: Please rephrase the sentence as it is not clear.

Author Response

RESPONSE LETTER

João Pessoa, PB, Brazil, December 04 2019

Ref.: ijerph-646056

            We appreciated the review of our manuscript and the suggestions given to make it suitable to International Journal of Environmental Research and Public Health high quality standards.

            As requested, this letter responds to each point raised by the reviewers. We inform that we have also prepared an individual letter for each reviewer.

            Please, kindly note that changes were tracked-up in the manuscript and are transcribed herein:

Reviewer #1

We appreciate the reviewer’s comments, which encouraged us to keep studying oral mucositis, a clinically relevant comorbidity afflicting children and adolescents with cancer. We have incorporated all suggestions into the manuscript and thoroughly revised the text as per request.

Line 73: Please change the number 9 that in superscript with square brackets.

Response: We apologize for our mistake and inform that the number 9 has been corrected accordingly.

Line 73: Regarding the 8 items of OAG please explain better to which areas they refer.

Response: We thank the reviewer for the scrutiny. The 8 items of OAG were explained and this information has been included in the sentence.

Line 75: Please write the correct definition ad "Cohen's Kappa.

Response: We inform that the correct definition was included.

Statistical analysis: Please provide a power analysis in which you explain how you decided to enroll a specific number of patients;

Response:

We thank the reviewer for the suggestion and clarify that we do not calculate the power of the study for the following reasons:

a) the sampling technique used was for convenience sampling, considering that childhood cancer is a rare condition and the authors point out the difficulties of including and accompanying children and adolescents with cancer in clinical studies (Russo, C. et al. Barriers and facilitators of clinical trial enrollment in a network of community-based pediatric oncology clinics. Pediatr Blood Cancer. 2019 Sep 25:e28023. doi: 10.1002/pbc.28023).

b) considering the particularities involving studies with cancer patients, especially pediatric patients, studies conducted in a single cancer center, there is a practical difficulty in obtaining a sufficient number of participants due to the low incidence of childhood câncer (Cheng, KK, Chang AM, Yuen MP. Prevention of oral mucositis in pediatric patients treated with chemotherapy: a randomised crossover trial comparing two protocols of oral care. Eur. J. Cancer. 2004;40(8):1208–1216. doi: 10.1016/j.ejca.2003.10.023). c) The sample consisted of 142 children. This number of participants is high compared to other studies with longitudinal design involving pediatric cancer patients for the purpose of studying mucositis in these patients. We emphasize that there are few published studies with this theme and epidemiological design developed in children with cancer.

 For example:

Allen G et al. The Prevalence and Investigation of Risk Factors of Oral Mucositis in a Pediatric Oncology Inpatient Population; a Prospective Study. J Pediatr Hematol Oncol. 2018 Jan;40(1):15-21. doi: 10.1097/MPH.0000000000000970. The study sample included 73 children. Velten DB et al.  Prevalence of oral manifestations in children and adolescents with cancer submitted to chemotherapy. BMC Oral Health. 2017 Jan 20;17(1):49. doi: 10.1186/s12903-016-0331-8. The study sample included 45 children aged months to 18 years. Yavuz B, Bal Yılmaz H. Investigation of the effects of planned mouth care education on the degree of oral mucositis in pediatric oncology patients. J Pediatr Oncol Nurs. 2015 Jan-Feb;32(1):47-56. doi: 10.1177/1043454214554011.  The study sample included 16 children aged 8 to 18 years.

Line 80-82: Please rephrase the sentence as it is not clear

Response: We inform that the sentence was rewritten.

Line 131-133: Why did you choose not to consider separately leukemias and lymphomas? I think this could really affect your results.

Response: This is a relevant point. Here, we provide some clarification as for not analyzing leukemias and lymphomas separately:

- all included patients were undergoing treatment in the same hospital;

- the protocols are standardized in a way that we can obtain accurate information on the chemotherapy treatments administered and corresponding dosage;

- although we not to consider separately leukemias and lymphomas, the most representative hematologic malignant was Acute Lymphoid Leukemia (ALL).

- The chemotherapy protocols adopted at Napoleão Laureano Hospital for ALL and lymphomas use methotrexate, which is a drug known to be stomatotoxic.

- In the literature is communn the classification - hematological malignancies and solid tumors. The guidelines of Multinational Association of Supportive Care in Cancer (MASCC) and European Society For Medical Oncology (ESMO) about hematological malignancies and their studies to determine predictive factors for complication in cancer patients include leukemias and lymphomas in the same group.

- Nevertheless, we agree with reviewer’s comment regarding the fact that not to consider separately leukemias and lymphomas could affect your results and we have included such an aspect as a limitation of the present study.

Line 134-135: Please add appropriate reference about the multidisciplinary and crucial role of dentists in helping the diagnosis of hematological head and neck malignancies (for your convenience: Patini et al. Early-stage diffuse large B-cell lymphoma of the submental region: a case report and review of the literature. Oral Surg. 2017;10:56-60).

Response: We thanks the reviewer to the suggestion. The suggested article is really  interesting, however, inform that we include  another study most directly related to this work's subject (13-Ribeiro et al. Differences between the oral changes presented by patients with solid and hematologic tumors during the chemotherapeutic treatment J Appl Oral Sci, 2020;28:e20190020; http://dx.doi.org/10.1590/1678-7757-2019-0020).

Line 205: Please rephrase the sentence as it is not clear.

Response: We thank the reviewer for the suggestion and inform that the sentence was rewritten.

With this, we believe that all requests were answered, and the present manuscript shall have the merit to be considered for publication in such an outstanding journal as International Journal of Environmental Research and Public Health.

Looking forward to hearing from you,

The authors.

Reviewer 2 Report

Authors sought to determine the variables that influence the time to occurrence of severe oral mucositis (SOM) in pediatric cancer patients receiving treatment. They adopted a longitudinal and observational study design and collected data from pediatric patients aged 0 – 19 years. Authors reported from their study that the duration to SOM occurrence was dependent on associated factors and the type of cancer.

COMMENTS:

The “survival analysis” included in the manuscript title appears misplaced and off-point, as article’s focus was merely on time to occurrence and no assessment was made on either disease free survival or overall survival. Worthy of mention also is that other studies have described similar findings for remission time. These studies include:

“Damascena LCL et al. Int. J. Environ. Res. Public Health 2018, 15(16):1153”

“Otmani N et al. Int. J. Paediatric Dentistry 2011; 21:210-216”

Line 90: Authors should do a better job of providing some descriptive content of the salient points presented in Table 1, rather than simply referring readers solely to the table.

Authors in line 94-95 only reported the overall SOM occurrence in their entire sample. What percentage of patients within the distinct hematologic and solid cancer subtypes groups developed SOM in their data set?

Authors should provide what the solid and dotted lines represent either within the graph (Figure 1 & 2) or include it as part of their figure legend. It is understood that the solid line represents the SOM occurrence time plot, while the dotted line indicates the confidence interval.

Article presented SOM occurrence time for the two cancer subgroups as median values and asserted that the median of hematologic malignancies (35.3 days) was smaller than that of solid tumors (77.1 days). While this may be true, the authors presented no indication nor reported if there was any significant difference between these two values (no p-value provided). This becomes important as the spread of the SOM occurrence data within both groups are large and well overlap.

More so, the discussion statements asserted in line 125 – 131 should be stated with caution because the referenced article “[10]” utilized mean as its measure of central tendency whereas this manuscript utilized median.

Author Response

RESPONSE LETTER

João Pessoa, PB, Brazil, December 04 2019

Ref.: ijerph-646056

            We appreciated the review of our manuscript and the suggestions given to make it suitable to International Journal of Environmental Research and Public Health high quality standards.

            As requested, this letter responds to each point raised by the reviewers. We inform that we have also prepared an individual letter for each reviewer.

            Please, kindly note that changes were tracked-up in the manuscript and are transcribed herein:

Reviewer #2 

The “survival analysis” included in the manuscript title appears misplaced and off-point, as article’s focus was merely on time to occurrence and no assessment was made on either disease free survival or overall survival. Worthy of mention also is that other studies have described similar findings for remission time. These studies include: “Damascena LCL et al. Int. J. Environ. Res. Public Health2018, 15(16):1153”; “Otmani N et al. Int. J. Paediatric Dentistry 2011; 21:210-216.

Response: We thank the reviewer and we provide some clarification as for Survival Analysis:

The survival analysis techniques are used when the study object variable is the time until the occurrence of an event of interest. Additionally, the analysis can add censored observations that are common in this type of study. The survival analysis methods can be used when the event of interest is death or remission, but applies equally to other outcomes (Collett D., Modeling Survival Data in Medical Research, 2003, Chapman & Hall). In this research, oral mucositis is the event of interest and from the survival analysis techniques it was possible to add the accurate information. In "Otmani N et al. Int. J. Paediatric Dentistry 2011; 21: 210-216" study, descriptive statistics and hypothesis tests to evaluate the incidence and determining factors of oral mucositis were used. The event of interest occurred for all patients and therefore there was no censored information.

Line 90: Authors should do a better job of providing some descriptive content of the salient points presented in Table 1, rather than simply referring readers solely to the table.

Response: We thank the reviewer for the suggestion and inform that a descrition of the salient points presented in Table 1 were included in the text.

Authors in line 94-95 only reported the overall SOM occurrence in their entire sample. What percentage of patients within the distinct hematologic and solid cancer subtypes groups developed SOM in their data set?

Response: We thank the reviewer for the suggestion. Information about the percentage of patients within the distinct hematologic and solid cancer subtypes groups developed SOM was included.

- Authors should provide what the solid and dotted lines represent either within the graph (Figure 1 & 2) or include it as part of their figure legend. It is understood that the solid line represents the SOM occurrence time plot, while the dotted line indicates the confidence interval.

Response: We appreciate the suggestion and inform that this information was included in the legends of figure 1 and figure 2.

Article presented SOM occurrence time for the two cancer subgroups as median values and asserted that the median of hematologic malignancies (35.3 days) was smaller than that of solid tumors (77.1 days). While this may be true, the authors presented no indication nor reported if there was any significant difference between these two values (no p-value provided). This becomes important as the spread of the SOM occurrence data within both groups are large and well overlap.

Response: We thank the reviewer for the suggestion and inform that this information was included in the outcomes.

More so, the discussion statements asserted in line 125 – 131 should be stated with caution because the referenced article “[10]” utilized mean as its measure of central tendency whereas this manuscript utilized median.

Response: We appreciate the reviewer’s scrutiny and inform that additional comments about this point have been included and discussed in the manuscript.

 With this, we believe that all requests were answered, and the present manuscript shall have the merit to be considered for publication in such an outstanding journal as International Journal of Environmental Research and Public Health.

Looking forward to hearing from you,

The authors.

Round 2

Reviewer 1 Report

The authors have not given acceptable reasons for the lack of a power analysis for the calculation of the sample size. Methodological and statistical consistency is not a trivial element in a scientific publication. If it is true what the authors assert in paragraph a) of their answer to my question on power analysis then the manuscript must be considered a "pilot study" and this definition must be inserted both in the title and in the materials and methods but, especially for what the authors themselves report in paragraph b) of the response to the reviewer, this is not obviously the case. Precisely in paragraph b) the same authors report that there are studies that have enrolled a smaller number of subjects. Starting from this evidence the authors are obliged to analyze the data of the previous articles, check if in those cases statistically significant differences were detected and use the values ​​of mean, standard deviation, alpha and beta error to correctly calculate the sample size. It is clear that if the previous studies have found statistically significant differences with a numerically lower sample also the authors of this article will find significant differences but this data does not exempt from precisely reporting the power analysis “in extenso”.

With regard to the separation between leukemias and lymphomas, please report in the manuscript what you wrote in the response to the reviewer on the guidelines of the Multinational Association of Supportive Care in Cancer (MASCC) and the European Society for Medical Oncology (ESMO) about hematological malignancies since it seems the only motivation that can be shared. The others are not methodologically acceptable.

I insist on suggesting to the authors that, in a journal that deals with cross-cutting issues such as public health, an intelligent way of attracting readers of various medical specializations is to prefer citations (such as the one I suggested in my previous report) that may also interest the dentists in their multidisciplinary role of guardians of oral health as a mirror for general health mantainance. I therefore invite you to reconsider your opinion with regard to the suggested citation, also remembering that the self-citations should normally be avoided (when not strictly necessary, but obviously this is not the case since the citation you inserted with number 13 was not present in the first draft of the manuscript).

In the hope that my concerns can be resolved.

Regards

Author Response

João Pessoa, PB, Brazil, December 13th, 2019

Ref.: ijerph-646056

We appreciate the scrutiny and attention given to our responses to the reviewers and inform that we have made an effort to substantiate hereby each point raised by the reviewers.

The celerity with which we replied to reviewers was due to the approaching deadline of the funding call for article processing charges sponsored by the Federal University of Paraíba in Brazil. More importantly, we would like to note that we were willing to revise the manuscript in accordance with all valuable suggestions and recommendation of the reviewers.

As per request, please kindly find enclosed the responses to all points raised by the reviewers. The changes were incorporated into the manuscript and are transcribed as follows:

Reviewer

1. The authors have not given acceptable reasons for the lack of a power analysis for the calculation of the sample size. Methodological and statistical consistency is not a trivial element in a scientific publication. If it is true what the authors assert in paragraph a) of their answer to my question on power analysis then the manuscript must be considered a "pilot study" and this definition must be inserted both in the title and in the materials and methods but, especially for what the authors themselves report in paragraph b) of the response to the reviewer, this is not obviously the case. Precisely in paragraph b) the same authors report that there are studies that have enrolled a smaller number of subjects. Starting from this evidence the authors are obliged to analyze the data of the previous articles, check if in those cases statistically significant differences were detected and use the values of mean, standard deviation, alpha and beta error to correctly calculate the sample size. It is clear that if the previous studies have found statistically significant differences with a numerically lower sample also the authors of this article will find significant differences but this data does not exempt from precisely reporting the power analysis “in extenso”.

Response: We experienced difficulties in establishing a sample size planning for this study because of the sample characteristics. Childhood cancer is a rare condition, which renders sample calculation a more complex task. In our study, participants were recruited through non-probabilistic convenience sampling for longitudinal and observational analysis based on the hospital’s demands. As requested by the reviewer, an a posteriori analysis was performed, which indicated statistical power of 77%, considering an alpha error of 5% and n = 142. Please, note that this information has been included in the Methods section.

With regard to the separation between leukemias and lymphomas, please report in the manuscript what you wrote in the response to the reviewer on the guidelines of the Multinational Association of Supportive Care in Cancer (MASCC) and the European Society for Medical Oncology (ESMO) about hematological malignancies since it seems the only motivation that can be shared. The others are not methodologically acceptable. 

Resposta: Agradecemos a sugestão e compreendemos que este ponto se torna relevante dentro do nosso estudo e incluímos esta explicação no artigo.

Response: We appreciate the suggestion and agree with the reviewer. The manuscript has been revised accordingly.

I insist on suggesting to the authors that, in a journal that deals with cross-cutting issues such as public health, an intelligent way of attracting readers of various medical specializations is to prefer citations (such as the one I suggested in my previous report) that may also interest the dentists in their multidisciplinary role of guardians of oral health as a mirror for general health mantainance. I therefore invite you to reconsider your opinion with regard to the suggested citation, also remembering that the self-citations should normally be avoided (when not strictly necessary, but obviously this is not the case since the citation you inserted with number 13 was not present in the first draft of the manuscript).

Response: We share the reviewer’s concern about including citations that consider the multidisciplinary role of dentists. Please, note that the suggested citation has been incorporated into the manuscript.

As for reference #13 (Ribeiro et al, 2019), we would like to clarify that such article had not been mentioned in the first version of the manuscript because it had not been published yet. This article was published on November 25th, 2019, and is available from http://www.scielo.br/scielo.php?script=sci_arttext&pid=S1678-77572020000100401&tlng=en

We reinforce that the article was cited because it is one of the few studies using the OAG tool in pediatric patients with cancer. The occurrence of mucositis over time was analyzed based on the diagnosis of solid and hematologic tumor in pediatric patients, which iss highly relevant for the context of the present study.

We appreciate the reviewer’s concerns and apologize for any misunderstanding in our communication. We hope that all requests were answered, and the present manuscript shall have the merit to be considered for publication in such an outstanding journal as International Journal of Environmental Research and Public Health.

Looking forward to hearing from you,

The authors.

Round 3

Reviewer 1 Report

The authors addressed quite satisfactorily my comments so the manuscript can be published in its present form.

Regards